# Rapid Quantification of *Salmonella* Typhimurium in Ground Chicken Using Immunomagnetic Chemiluminescent Assay

**DOI:** 10.3390/microorganisms13040871

**Published:** 2025-04-10

**Authors:** Sandhya Thapa, Niraj Ghimire, Fur-Chi Chen

**Affiliations:** 1Department of Food and Animal Sciences, Tennessee State University, Nashville, TN 37209, USA; sthapa1@tnstate.edu (S.T.);; 2Department of Human Sciences, Tennessee State University, Nashville, TN 37209, USA

**Keywords:** *Salmonella* Typhimurium, ground chicken, chemiluminescent, immunomagnetic, quantification, monoclonal antibodies

## Abstract

Many countries have established regulatory frameworks to monitor and mitigate *Salmonella* contamination in poultry products. The ability to rapidly quantify *Salmonella* is critical for poultry processors to facilitate early detection, implement corrective measures, and enhance product safety. This study aimed to develop an Immunomagnetic Chemiluminescent Assay (IMCA) for the quantification of *Salmonella* Typhimurium in ground chicken. Immunomagnetic microbeads functionalized with monoclonal antibodies were employed to selectively capture and concentrate *Salmonella* from ground chicken samples. A biotin-labeled monoclonal antibody, followed by an avidin-horseradish peroxidase conjugate, was used to bind the captured bacteria and initiate a chemiluminescent reaction catalyzed by peroxidase. Light emission was quantified in relative light units (RLUs) using two luminometers. Ground chicken samples were inoculated with a four-strain S. Typhimurium cocktail ranging from 0 to 3.5 Log CFU/g. Bacterial concentrations were confirmed using the Most Probable Number (MPN) method. Samples underwent enrichment in Buffered Peptone Water (BPW) supplemented with BAX MP Supplement at 42 °C for 6 and 8 h before analysis via IMCA. A linear regression analysis demonstrated that the optimal quantification of *Salmonella* was achieved at the 8 h enrichment period (R^2^ ≥ 0.89), as compared to the 6 h enrichment. The limit of quantification (LOQ) was determined to be below 1 CFU/g. A strong positive correlation (R^2^ ≥ 0.88) was observed between IMCA and MPN results, indicating methodological consistency. These findings support the application of IMCA as a rapid and reliable method for the detection and quantification of *Salmonella* in ground chicken.

## 1. Introduction

Belonging to the class Gammaproteobacteria and the family Enterobacteriaceae, *Salmonella* is a genus of rod-shaped, Gram-negative, non-spore-forming bacteria that are facultatively anaerobic, chemo-organotrophic, and typically motile [1,2]. Non-typhoidal *Salmonella* species are widely recognized as the predominant bacterial pathogens responsible for foodborne illness [3], accounting for approximately 11% of all cases, 35% of hospitalizations, and 28% of deaths associated with foodborne infections [4,5]. According to the Centers for Disease Control and Prevention (CDC), *Salmonella* is estimated to cause over 1.35 million infections annually in the United States, leading to approximately 26,500 hospitalizations and 420 fatalities [6]. The economic burden associated with medical costs, productivity losses, and premature mortality due to foodborne *Salmonella* infections in the United States is estimated to range between USD 4 billion and USD 11 billion per year [7].

The genus *Salmonella* comprises two species, *Salmonella* enterica and *Salmonella* bongori, which collectively exhibit over 2600 distinct serovars [8,9]. These bacteria demonstrate remarkable adaptability, enabling their persistence across a diverse range of animal hosts, including humans [10]. Among the serovars associated with foodborne illness, *Salmonella* enterica subsp. enterica serovar Typhimurium ranks as the second most prevalent, following *Salmonella* enterica subsp. enterica serovar Enteritidis [11]. *Salmonella* contamination has been documented across a broad spectrum of food products, including fresh produce, meat, and poultry [12]. However, poultry products represent the primary vector for human infections [13]. The Interagency Food Safety Analytics Collaboration (2022) reported that poultry consumption was implicated in more than 23% of foodborne *Salmonella* infections, with chicken alone accounting for over 17% of cases [14]. Given that poultry is the most widely consumed meat in the United States, its potential to serve as a reservoir and transmission vehicle for foodborne pathogens presents a significant public health concern [15].

Various methodologies have been developed for the quantitative detection of *Salmonella*. Traditional culture-based approaches, such as the Standard Plate Count method, remain widely employed due to their cost-effectiveness and simplicity and are still considered the gold standard in numerous countries [16]. However, these methods rely on bacterial growth in selective culture media, necessitating prolonged incubation and labor-intensive colony isolation and biochemical identification procedures [17,18]. The initial detection of suspect colonies typically requires 2–3 days, whereas the confirmation of bacterial species may take over a week [19]. Moreover, the presence of viable but non-culturable cells may compromise sensitivity, leading to false-negative results [17,20].

To overcome the limitations associated with conventional culture-based techniques, immunological assays such as enzyme-linked immunosorbent assay (ELISA) and lateral flow immunoassays have been developed for the rapid quantitative detection of *Salmonella* Typhimurium [21,22]. These methods offer advantages in terms of simplicity, stability, and reduced detection time. However, their performance is often constrained by the necessity for pre-enrichment steps, complex sample pretreatment, limited sensitivity, and the potential for false-negative results [23]. Consequently, the development of alternative detection methodologies remains imperative.

Polymerase chain reaction (PCR)-based molecular assays have been extensively employed for *Salmonella* detection [24,25] due to their superior speed and sensitivity compared to immunological methods. However, PCR-based techniques present several challenges, including automation difficulties, susceptibility to false-negative results due to PCR inhibitors within samples, the requirement for DNA purification, challenges in distinguishing viable from non-viable cells, reliance on accurate primer design, risks of cross-contamination, and the necessity for trained personnel [26,27,28]. Similarly, Raman spectroscopy has emerged as a promising, non-destructive analytical technique for bacterial detection. However, its applicability is hindered by the requirement for specialized instrumentation and expertise.

Biosensors are increasingly gaining traction due to their portability, compact design, and capability for multiplex detection. However, they are subject to challenges such as electrode interference and instability of nanomaterials used in sensor fabrication [16,29]. The Most Probable Number (MPN) method is another widely employed approach for bacterial quantification [18]. This technique is based on serial 10-fold dilutions and is commonly used for bacterial enumeration [30,31,32]. Despite its efficacy, MPN is labor-intensive, expensive, and time-consuming [4], highlighting the need for more rapid, reliable, and cost-effective detection methods.

An Immunomagnetic Chemiluminescent Assay (IMCA) integrates immunomagnetic separation and chemiluminescence principles to detect and quantify specific analytes, such as bacterial antigens and antibodies. The primary advantage of immunomagnetic bead-based assays lies in their enhanced antigen-binding efficiency, owing to a larger immobilization surface area. The incorporation of magnetic beads into chemiluminescence immunoassays offers distinct advantages, including increased sensitivity, selectivity, and reproducibility, while also facilitating streamlined and automated workflows [33]. IMCA allows for the efficient integration of key analytical steps—including reagent addition, sample dilution, incubation, washing, and signal measurement—into a cohesive and automated process [34]. This technique holds promise as a rapid and efficient tool for the detection of foodborne pathogens in food processing environments and regulatory inspection laboratories [35].

Numerous studies have demonstrated the applicability of immunomagnetic bead-based assays for the detection of toxins in agricultural commodities [33,36], precise quantification of plant hormones [37], and rapid identification of bacterial pathogens [35,38]. Additionally, immunomagnetic beads have been utilized in clinical settings for the sensitive detection of carcinoembryonic antigen [39,40]. However, relatively few studies have explored their application for the detection and quantification of foodborne bacterial pathogens [35,38]. Therefore, the objective of this study was to develop an IMCA for the quantitative detection of *Salmonella* Typhimurium in ground chicken.

## 2. Materials and Methods

### 2.1. Materials and Instruments

Four distinct strains of *Salmonella* Typhimurium (ATCC 13311, 29629, 49416, 59812) were obtained from the American Type Culture Collection (ATCC), Manassas, VA, USA, and stored at −80 °C until use. Ground chicken packages utilized in the experiment were purchased from a local grocery store.

Tryptic Soy Agar (TSA), Tryptic Soy Broth (TSB), Xylose-Lysine-Tergitol 4 (XLT-4) Agar, and Buffered Peptone Water (BPW) were sourced from Thermo Fisher Scientific Inc., Lenexa, KS, USA, while MP media was acquired from Hygiena, Camarillo, CA, USA. Bovine Serum Albumin (BSA), 10X Phosphate-Buffered Saline (PBS), and TWEEN 20 were obtained from Fisher Scientific, Fair Lawn, NJ, USA. A 1X PBS solution containing 0.05% TWEEN 20 (PBST) was prepared in the laboratory from 10X PBS and used as a working buffer.

The Hula mixer, Dynal magnet, and Dynabeads antibody coupling kit were purchased from Invitrogen, Carlsbad, CA, USA. SystemSURE, BAX^®^ Q7 System, Q swabs, and *Salmonella* Quant kits were obtained from Hygiena, Camarillo, CA, USA. SuperSignalTM ELISA Femto Luminol Enhancer Solution and SuperSignalTM ELISA Femto Stable Peroxide Solution were procured from Thermo Fisher Scientific, Rockford, IL, USA, while the GloMAX 20/20 luminometer was acquired from Promega Corporation, Madison, WI, USA. Monoclonal antibodies (MAb-1E10 and MAb-1C8) were prepared in-house using flagellin from Salmonella Typhimurium (ATCC 13311). The specificity and epitope mapping of these monoclonal antibodies have been detailed in our previous studies [23,41].

### 2.2. Antibody-Coated Dynabeads and Working Solutions

Antibody-coupled Dynabeads were prepared following the protocol provided in the Dynabeads^®^ Antibody Coupling Kit (Invitrogen, Waltham, MA, USA). The antibody buffer consisted of 10 mg/mL BSA in PBST. Dynabeads were functionalized with MAb-1C8, while the biotin-labeled MAb-1E10 antibody was diluted in antibody buffer at a 1:10,000 ratio. Similarly, the Avidin–HRP conjugate was diluted in antibody buffer at the same ratio. The chemiluminescent substrate was prepared by mixing equal volumes of SuperSignalTM ELISA Femto Luminol Enhancer Solution and SuperSignalTM ELISA Femto Stable Peroxide Solution.

### 2.3. Preparation of Bacterial Cocktail

Four different strains of S. Typhimurium (ATCC 13311, 29629, 49416, 59812) were streaked from glycerol stocks onto TSA plates and incubated at 37 °C for 22–24 h. A single colony from each strain was individually inoculated into 10 mL of TSB and incubated overnight at 37 °C. Equal volumes of each TSB culture were combined to form a bacterial cocktail. The *Salmonella* concentration in the cocktail was determined using the Plate Count (PC) method through 10-fold serial dilutions, plated in duplicate on TSA and XLT-4 agar. Viable cells were quantified as log colony-forming units per gram (Log CFU/g). TSA counts were used to determine inoculation levels in ground chicken samples, while XLT-4 was employed to check for potential bacterial contamination in *Salmonella* cultures.

### 2.4. Preparation of Positive Control

A single colony of S. Typhimurium (ATCC 13311) from a TSA plate was inoculated into a filter bag containing 225 mL of BPW and homogenized for 30 s using a Stomacher Circulator to ensure uniform bacterial distribution. The homogenate was incubated at 37 °C for 24 h, after which 1 mL of the mixture was aliquoted into Eppendorf tubes and stored at −80 °C for use as positive control in subsequent experiments.

### 2.5. Ground Chicken Samples and Enrichment Protocol

Ground chicken samples (32.5 g) in duplicate were artificially contaminated at various target levels ranging from −0.5 to 3.5 log CFU/g to evaluate contamination levels. Uninoculated samples were included as negative controls. The negative controls were further confirmed to be free of *Salmonella* using BAX System Real-Time PCR Assay. The protocol recommends mixing 325 g of sample to 1625 mL BPW, and only 30 mL of homogenate was transferred to a sterile filtered bag. Subsequently, 30 mL of 42 °C prewarmed MP media containing 40 mg/L novobiocin was added, and the samples were incubated at 42 °C for 6 and 8 h. In this study, considering the efficiency of disposal treatment of the contaminated materials, proportionally reduced samples (32.5 g) were mixed in sterile filter bags containing 162.5 mL of BPW. The subsequent procedures remained the same.

### 2.6. IMCA of Enriched Ground Chicken Samples

The enriched cultures from ground chicken samples were collected at 6 and 8 h for analysis. A 900 μL aliquot of enriched sample culture was mixed with 100 μL of 10X PBST and heated for 10 min. The sample was subsequently centrifuged for 10 min, and 500 μL of the supernatant was carefully collected. MAb-1C8 coupled Dynabeads (5 μL) were added and incubated in a Hula mixer for 15 min, followed by magnetic separation using the Dynal magnet. The sample was subjected to two washing steps with 1X PBST, with supernatant removal at each step. Biotin-labeled MAb-1E10 (500 μL) was added and incubated for 15 min, followed by additional magnetic separation and washing. Finally, the Avidin–HRP conjugate (500 μL) was added, and after a final washing step, 100 μL of chemiluminescent substrate was added. A 20 μL aliquot was analyzed using the GloMAX luminometer (Promega, Madison, WI, USA) while another 20 μL aliquot was tested using SystemSURE (Hygiena, Camarillo, CA, USA). A schematic representation of the IMCA procedure is depicted in Figure 1.

### 2.7. MPN and Real-Time PCR Assay

BPW homogenates from Section 2.5 were manually mixed for 30 s, and MPN sets were prepared in triplicate using five serial dilutions (1, 0.1, 0.01, 0.001, and 0.0001 g). The first set of tubes contained 4 mL of BPW, to which 6 mL of homogenate was added. Subsequent serial dilutions were prepared by transferring 1 mL from the preceding dilution into 9 mL of BPW. Tubes were incubated at 37 °C for 24 h. *Salmonella* presence was confirmed using the BAX System Real-Time PCR Assay, and MPN values were determined using the MLG Appendix 2.05 MPN table, subsequently converted into Log MPN/g.

### 2.8. Data Analysis

A completely randomized design was employed, with the experiment replicated 12 times under identical conditions (Figure 2). Each inoculation level was tested in triplicate per experiment. Data from replicates were averaged to minimize variability and enhance reliability. Contamination levels (Log CFU/g) and IMCA responses (relative light units, RLUs) from GloMAX and SystemSURE were expressed as percentage positive responses (%*p*) relative to the positive control. A linear regression model was developed for 6 and 8 h enrichment times by plotting contamination levels against %*p*, with R-squared values calculated to determine the best-fitting regression curve. Statistical analyses were performed using Microsoft Excel (Microsoft 365, 2021) and SAS (version 9.4, SAS Institute, Cary, NC, USA).

## 3. Results

### 3.1. Culture Enumeration for Preparing Contaminated Ground Chicken Samples

Inoculums were initially prepared by diluting the culture, and ground chicken samples were inoculated with varying inoculum concentrations as specified in the experimental design. The enumeration of *Salmonella* cultures in tryptic soy broth (TSB) was conducted using two plate count methods: tryptic soy agar (TSA) and xylose lysine tergitol 4 agar (XLT-4). The mean *Salmonella* count on TSA was 8.90 Log CFU/mL with a standard deviation (SD) of 0.11 Log CFU/mL, while the mean count on XLT-4 was 8.92 Log CFU/mL with an SD of 0.11 Log CFU/mL across 12 experimental sets.

A statistical analysis revealed no significant difference (*p* > 0.05) between TSA and XLT-4 within the same experimental set, indicating that both methods were equally effective for *Salmonella* enumeration under the conditions of this study (Figure 3). However, a significant difference (*p* < 0.05) was observed among different experimental sets under similar conditions, which may be attributed to the variability introduced by loop inoculation of colonies into TSB from stock plates.

For the purposes of this study, TSA counts were used to determine contamination levels in ground chicken samples, whereas XLT-4 was employed to assess potential contamination by non-*Salmonella* bacteria within the cultures. The consistency of TSA and XLT-4 within the same experimental set indicated the purity of the *Salmonella* culture.

### 3.2. Positive and Negative Controls in the IMCA

Considering the quantification ranges of both instruments, a positive control sample was prepared as described in Section 2.4, and the same positive control was utilized in each experimental set. A negative control was prepared from ground chicken samples devoid of inoculum for each experiment. The luminescence readings of both positive and negative controls across 12 independent experiments, obtained using the GloMAX and SystemSURE luminometers, are presented in Table 1.

For measurements using the GloMAX luminometer, the mean luminescence of the positive controls was 1.39 × 10⁹ RLUs, with a standard deviation (SD) of 2.04 × 10⁸ RLUs, whereas the mean luminescence of the negative controls was 7.11 × 10⁴ RLUs, with an SD of 5.76 × 10⁴ RLUs. In contrast, measurements obtained using the SystemSURE luminometer (Hygiena, Camarillo, CA, US) yielded a mean luminescence of 8.38 × 10^3^ RLUs for positive controls (SD = 1.68 × 10^2^ RLUs) and 1.3 RLUs for negative controls (SD = 1.3 RLUs).

These results indicate significant differences in the sensitivity and dynamic range of the two instruments. The GloMAX luminometer is a highly sensitive device optimized for precise bioluminescence measurements, whereas the SystemSURE luminometer is a portable, handheld device designed primarily for ATP assays and was adapted for this study using a swab-based format. Although both instruments report luminescence in relative light units (RLUs), it is important to note that RLU values are not standardized across different luminometers. Consequently, direct comparisons of RLU values between the GloMAX and SystemSURE instruments are not valid.

### 3.3. GloMAX and SystemSURE Response Curves of Two Enrichment Conditions

Since relative light units (RLUs) are instrument-specific measurements, they are inherently dependent on the assay and detection system used. Consequently, although both instruments measure RLUs, their values are not directly comparable due to variations in sensor technology and calibration methods. To facilitate a comparative analysis, the RLU values for each sample were normalized to a percentage positive response (%*p*), defined as %*p* = (RLU of sample/RLU of positive control) × 100.

The efficacy of the GloMAX and SystemSURE luminometers for quantifying *Salmonella* Typhimurium in ground chicken was assessed using two different enrichment durations (6 and 8 h). Linear regression models were generated for both enrichment time points to determine the optimal duration for quantification. Correlations between contamination levels (Log CFU/g) and the percentage positive response (%*p*) obtained from GloMAX at both enrichment time points are presented in Figure 4. For the 8 h enriched ground chicken samples, a linear detection range of −0.7 to 3.5 Log CFU/g was established (R^2^ ≥ 0.89). In contrast, the 6 h enrichment resulted in a narrower linear range of 1.5 to 3.5 Log CFU/g (R^2^ ≥ 0.80). The broader linear range observed with the 8 h enrichment (−0.7 to 3.5 Log CFU/g) suggests an enhanced ability to quantify lower contamination levels compared to the 6 h enrichment (1.5 to 3.5 Log CFU/g).

Similarly, correlations between contamination levels (Log CFU/g) and the positive percentage response (%*p*) obtained from SystemSURE for both enrichment durations are presented in Figure 5. For the 8 h enrichment, a linear range of −0.7 to 1.7 Log CFU/g was observed (R^2^ ≥ 0.83), whereas the 6 h enrichment resulted in a detection range of 1.5 to 3.5 Log CFU/g (R^2^ ≥ 0.86). While SystemSURE effectively quantified contamination levels, its quantification range was narrower compared to GloMAX.

These results (Figure 4 and Figure 5) indicate that at 8 h of enrichment, both instruments could detect low contamination levels (−0.7 Log CFU/g). However, at 6 h, they were limited to quantifying higher contamination levels (≥1.5 Log CFU/g). Furthermore, GloMAX exhibited a broader quantification range than SystemSURE. These findings suggest that an 8 h enrichment period, particularly when using GloMAX (Promega, Madison, WI, USA), provides a more extensive detection range for quantifying *Salmonella* Typhimurium in the experimental setting.

### 3.4. Comparison of GloMax and MPN

To evaluate the performance of the proposed method in comparison with the Most Probable Number (MPN) method, a simple linear regression analysis was performed. Log MPN/g values were determined using the MPN table from MLG Appendix 2.05, while Log CFU/g values for GloMAX were calculated using the equation derived from the linear regression model of 8 h enriched ground chicken samples.

The analysis revealed a strong positive correlation (R^2^ ≥ 0.9) between contamination levels (Log CFU/g) and Log MPN/g (Figure 6). Similarly, a strong positive correlation (R^2^ ≥ 0.88) was observed between contamination levels (Log CFU/g) and Log CFU/g values obtained from GloMAX. These results demonstrate the consistency between the two quantification methods.

Further examination of the regression curve indicated that GloMAX provided a more accurate estimation of contamination levels compared to the MPN method. Specifically, MPN yielded values approximately 0.5 Log CFU/g higher than the actual contamination levels, suggesting a slight overestimation. However, no statistically significant difference was observed between the Log CFU/g estimates of the two methods.

These findings suggest that GloMAX serves as a reliable and effective alternative to the MPN method for the quantification of *Salmonella* Typhimurium in ground chicken samples. Like MPN, GloMAX is not intended for identification or serotyping of *Salmonella*. The monoclonal antibodies used in this study may exhibit cross-reactivity with some closely related serovars that have similar phase 1 (i) flagellin or phase 2 (1,2) flagellin. Further antigenic confirmation is required for the positive identification of these serovars.

## 4. Discussion

Tryptic Soy Agar (TSA) is capable of supporting the growth of *Salmonella*; however, it is non-selective and allows the proliferation of various bacterial species. In contrast, Xylose Lysine Tergitol-4 (XLT-4) agar contains ferric compounds that react with hydrogen sulfide (H_2_S) produced by *Salmonella* strains, resulting in the formation of black-colored colonies. Consequently, *Salmonella* colonies typically appear black or red with black centers on XLT-4, facilitating their identification. Additionally, XLT-4 incorporates tergitol, which inhibits the growth of Proteus species, Pseudomonas, and Gram-positive bacteria [42,43,44]. However, not all *Salmonella* strains exhibit equal growth efficiency on XLT-4.

A comparative analysis of *Salmonella* recovery between selective media and TSA under various stress conditions—such as nutrient deprivation, acid stress, desiccation, and thermal stress—revealed similar recovery rates [45]. In contrast, a significantly higher *Salmonella* recovery was observed on TSA compared to XLT-4 across a temperature range of 4–45 °C, suggesting that nutrient-limited environments, such as wastewater, impose stress that affects *Salmonella* growth on selective media like XLT-4 [46].

The limit of quantification (LOQ) refers to the minimum analyte concentration that can be accurately and precisely quantified under specified conditions [47]. It is typically calculated by determining the mean detection signal of the control sample and adding ten times the standard deviation of that control signal [48]. LOQ is a critical parameter for evaluating assay sensitivity and precision, particularly when detecting low bacterial concentrations. The ability to quantify *Salmonella* at low levels is essential for identifying potential contamination that might otherwise remain undetected, thereby ensuring consumer safety and regulatory compliance, as even minimal *Salmonella* contamination can lead to foodborne illness [30]. Several studies have focused on developing and validating quantification methods for *Salmonella* in poultry with lower LOQ values [4,49,50].

The findings of this study are consistent with previous research on *Salmonella* quantification, including the development of a rapid RT-PCR method for detecting *Salmonella* in pork and beef lymph nodes [5], an RT-PCR method for quantifying *Salmonella* in poultry [4], and a qPCR method for measuring *Salmonella* in the feces and tissues of sheep [18]

Among commonly used quantification methods, the Most Probable Number (MPN) and standard plate count methods are widely employed. While the MPN method is effective for detecting low *Salmonella* concentrations, it is both time-consuming and labor-intensive [4,18,51]. Additionally, when microbial concentrations are high, the MPN method tends to overestimate bacterial counts [4]. In contrast, the standard plate count method, which utilizes selective and differential media, is frequently used [18]. However, the process of isolating and biochemically identifying presumptive colonies is costly, time-intensive, and requires up to four days to obtain definitive results [18,52].

The GloMAX system presents a more efficient alternative, particularly in scenarios where rapid and accurate quantification is critical. The results of this study align with a previous study [4], where it was reported that *Salmonella* Quant provides faster and more reliable estimations compared to MPN, which tends to overestimate bacterial concentrations and requires a longer time for confirmatory results. Similarly, another study [5] demonstrated that the BAX^®^ System SalQuant^®^ yielded results comparable to the MPN method, suggesting that it represents a viable alternative for *Salmonella* quantification.

Recent studies have focused on assessing *Salmonella* prevalence and concentrations, leading to the development of various quantification methods for different food matrices [4,5,18,30,53,54,55,56,57,58]. In alignment with the Healthy People 2030 objectives, which aim to reduce human *Salmonella* infections by 25%, the USDA Food Safety and Inspection Service (FSIS) has introduced a new *Salmonella* framework designed to mitigate foodborne *Salmonella* cases associated with poultry products. This framework emphasizes the importance of improved quantification methods and includes a Statistical Process Control (SPC) program aimed at reducing aerobic bacterial counts by at least 1 log between two processing stages. These regulatory measures underscore the necessity of developing quantification techniques that are accurate, precise, rapid, reliable, and characterized by low limits of detection and quantification [5,59].

## 5. Conclusions

The findings of this study demonstrate the efficacy and accuracy of an Immunomagnetic Chemiluminescent Assay (IMCA) for the quantitative detection of *Salmonella* Typhimurium in ground chicken. The optimal enrichment time for an IMCA was determined to be 8 h. Notably, the MPN method was found to overestimate contamination levels compared to IMCA. Consequently, an IMCA presents a promising alternative for *Salmonella* quantification, offering a faster and more streamlined approach relative to MPN. An IMCA provides results within 11 h, including enrichment time, whereas the MPN method requires at least one day for BAX confirmation and up to a full week for complete verification.

Furthermore, the cost-effectiveness of IMCA enhances its suitability for *Salmonella* enumeration in ground chicken, making it an appealing alternative to conventional methods. This study underscores the significance of adopting rapid, efficient, and cost-effective quantification techniques such as an IMCA to facilitate timely and reliable *Salmonella* detection. By delivering results in less than a day, an IMCA not only improves testing efficiency but also contributes to the production of safer, higher-quality food products. Consequently, its implementation may play a critical role in mitigating public health risks associated with S. Typhimurium contamination in ground chicken.

## Figures and Tables

**Figure 1 microorganisms-13-00871-f001:**
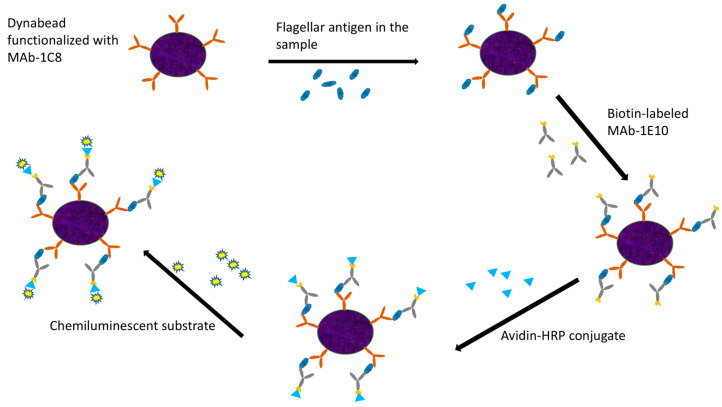
Graphic representation of the main steps of IMCA.

**Figure 2 microorganisms-13-00871-f002:**
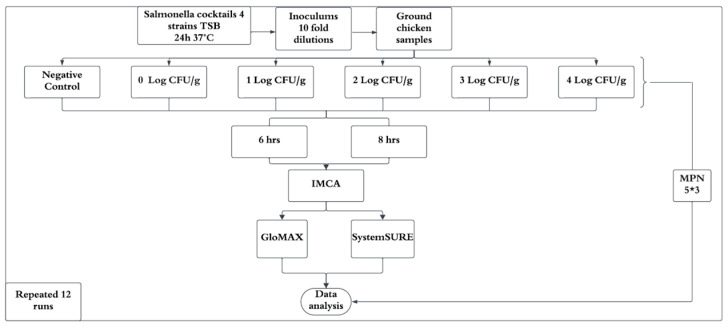
Overall experimental design.

**Figure 3 microorganisms-13-00871-f003:**
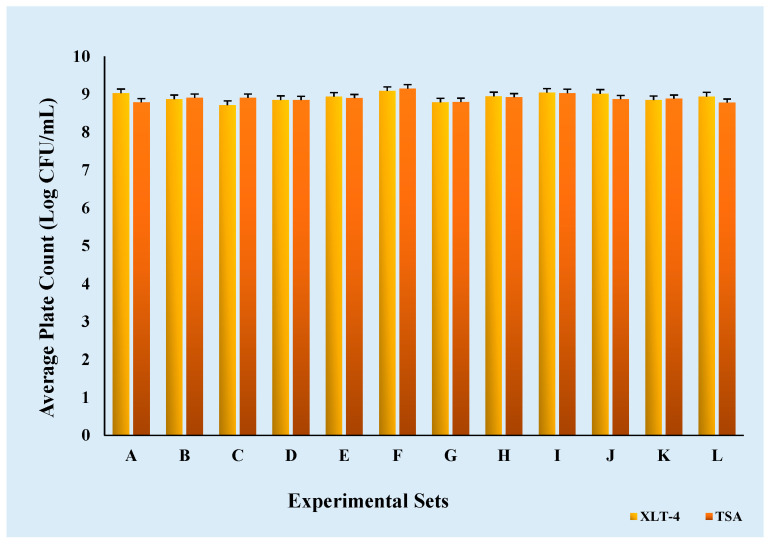
Comparison of TSA and XLT-4 agar for the enumeration of *Salmonella* cultures in 12 experimental sets (A-L).

**Figure 4 microorganisms-13-00871-f004:**
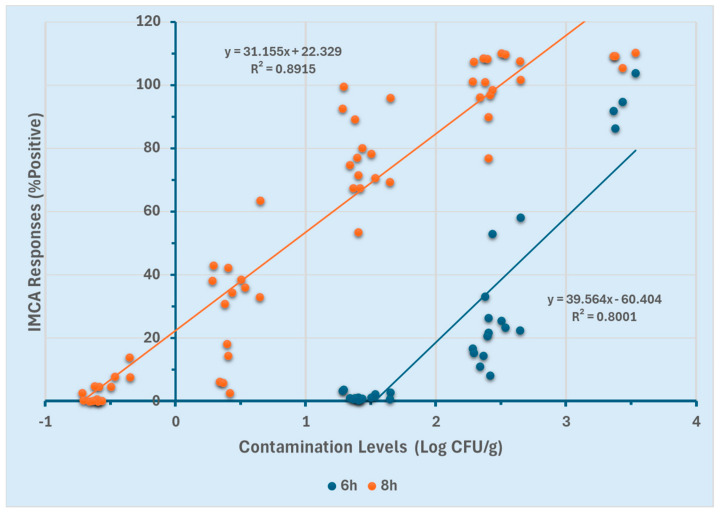
GloMAX response curves using two different enrichment durations (6 and 8 h).

**Figure 5 microorganisms-13-00871-f005:**
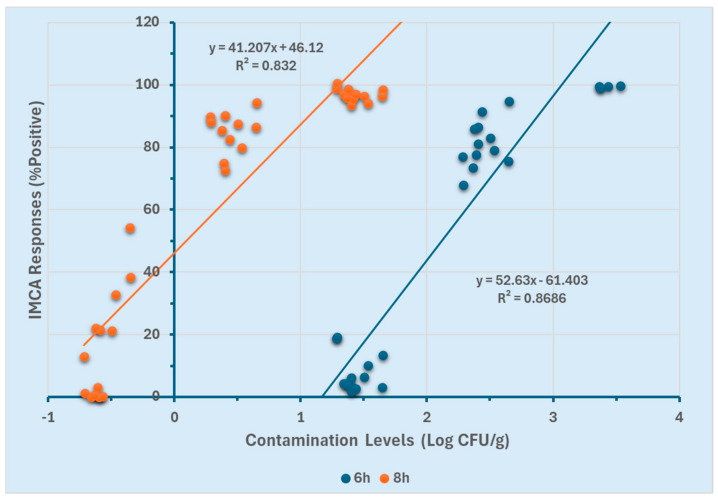
SystemSURE response curves using two different enrichment durations (6 and 8 h).

**Figure 6 microorganisms-13-00871-f006:**
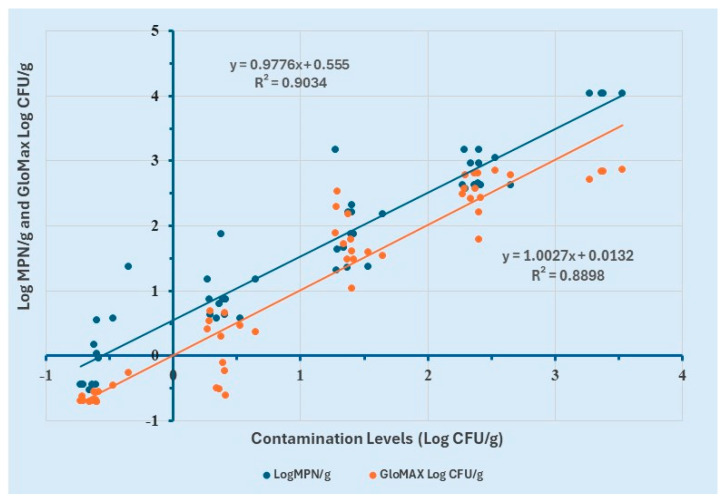
Comparison of GloMAX and MPN for the quantification of contamination levels of *Salmonella* Typhimurium in ground chicken.

**Table 1 microorganisms-13-00871-t001:** Descriptive analysis of positive and negative controls in 12 experimental sets.

	GloMAX		SystemSURE	
	Positive Control (RLUs)	Negative Control (RLUs)	Positive Control (RLUs)	Negative Control (RLUs)
Means	1.39 × 10^9^	7.11 × 10^4^	8.38 × 10^3^	1.3
SD	2.04 × 10^8^	5.76 × 10^4^	1.68 × 10^2^	1.3
Min	8.63 × 10^8^	6.75 × 10^3^	7.96 × 10^3^	0
Max	1.79 × 10^9^	2.29 × 10^5^	8.61 × 10^3^	5

## Data Availability

The original contributions presented in this study are included in the article. Further inquiries can be directed to the corresponding author.

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
