# Peer review of "Rapid Quantification of Salmonella Typhimurium in Ground Chicken Using Immunomagnetic Chemiluminescent Assay"

_microorganisms, 2025, doi:10.3390/microorganisms13040871_

Round 1
Reviewer 1 Report
Comments and Suggestions for Authors
I appreciate the opportunity to evaluate the manuscript titled "Rapid Quantification of Salmonella Typhimurium in Ground Chicken using Immunomagnetic Chemiluminescent Assay".
The research outlines a meticulously structured and conducted study of an immunomagnetic chemiluminescent technique aimed at quantifying Salmonella Typhimurium in ground chicken. The findings suggest that this approach may function as a fast and accurate alternative for conventional methods of Salmonella detection and quantification.
The manuscript topic is contemporary and pertinent, as it tackles a significant concern in food safety: rapid and precise identification of Salmonella in poultry products. The methodology is articulated comprehensively, encompassing the preparation of bacterial cultures, the IMCA procedure, and the Most Probable Number (MPN) method employed for comparative analysis. The inclusion of MPN enhances the robustness of the study, as it is a widely recognized and trustworthy reference method.
The researchers conducted comprehensive statistical evaluations, encompassing linear regression analysis, mean comparisons, and correlation assessments, to assess the efficacy of the IMCA method, which is adequate for deriving conclusions from the experimental results.
The findings are articulated with clarity and precision. The acknowledgment of the study's limitations by the authors is commendable, particularly regarding the variability introduced by loop inoculation and the discrepancies in sensitivity and dynamic range between the two luminometers employed.
Nevertheless, there are several minor issues that warrant attention, and I would be grateful if the authors could address them adequately to enhance the strength and scientific rigor of the work:
- In addition to the assertions made in Line 148 and Line 183, it seems that the study did not clarify whether the ground chicken samples underwent testing in duplicate or triplicate formats. This data is essential for evaluating the dependability and consistency of the findings.
- What factors could account for the ground chicken samples weighing 32.5 g? It is standard practice to utilize samples weighing 25 grams.
- Line 117. The investigation concentrated exclusively on ground chicken specimens acquired from a nearby grocery establishment. Although this establishes a controlled matrix, it may not entirely capture the variability inherent in ground chicken products derived from diverse sources or processing conditions. Additionally, did the authors test the purchased samples prior to conducting artificial inoculation to ensure that the samples are genuinely negative for Salmonella Typhimurium?
- Line 44. It is recommended to utilize "serovars" in place of "serotypes" to align with the established taxonomy nomenclature.
Reviewer 2 Report
Comments and Suggestions for Authors
Major Issue need to address,
Major Issue
- Lack of Specificity: The study does not address the specificity of the assay. Essential controls are missing, and the authors have not convinced me that this assay is specific to Salmonella typhimurium. They have not included closely related species such as S. Enteritidis, S. Heidelberg, S. Newport, S. Anatum, S. Infantis, S. Agona, S. Thompson, S. Saintpaul, S. Montevideo, and S. Muenchen. To improve method specificity, authous must include at least the top ten closely related serovars.
- Missing Non-Salmonella Controls: The authors must include non-Salmonella controls (e.g., E. coli, Enterococcus, Listeria, Campylobacter) to demonstrate that the detection is specific to Salmonella typhimurium.
- Subtypes Within Typhimurium: There are several subtypes within S. Typhimurium, including monophasic strains with the antigenic formula 1,4,[5],12:i:- and other variants. It is unclear whether the method works against all Typhimurium variants. These must be included in the study to ensure comprehensive validation.
Round 2
Reviewer 2 Report
Comments and Suggestions for Authors
Thank you for the clarification
Comments on the Quality of English LanguageThe revised version is good for publication.